# INA: An Integrative Approach for Enhancing Negotiation Strategies with Reward-Based Dialogue System

**Zishan Ahmad**[†], **Suman Saurabh**[†], **Vaishakh Sreekanth Menon**[†], **Asif Ekbal**[†],
**Roshni Ramnani**[‡], **Anutosh Maitra**[‡]

[†]Department of Computer Science and Engineering
Indian Institute of Technology Patna, India
{zeeman.zishan, saurabhm.SS89, vaishakhsm, asif.ekbal}@gmail.com
[‡]Accenture Labs, Bangalore, India
{roshni.r.ramnani, anutosh.maitra}@accenture.com

## Abstract

In this paper, we propose a novel negotiation dialogue agent designed for the online marketplace. Our agent is integrative in nature *i.e*, it possesses the capability to negotiate on price as well as other factors, such as the addition or removal of items from a deal bundle, thereby offering a more flexible and comprehensive negotiation experience. We create a new dataset called **Integrative Negotiation Dataset (IND)** to enable this functionality. For this dataset creation, we introduce a new semi-automated data creation method, which combines defining negotiation intents, actions, and intent-action simulation between users and the agent to generate potential dialogue flows. Finally, the prompting of GPT-J, a state-of-the-art language model, is done to generate dialogues for a given intent, with a human-in-the-loop process for post-editing and refining minor errors to ensure high data quality. We employ a set of novel rewards, specifically tailored for the negotiation task to train our Negotiation Agent, termed as the **Integrative Negotiation Agent (INA)**. These rewards incentivize the chatbot to learn effective negotiation strategies that can adapt to various contextual requirements and price proposals. By leveraging the IND, we train our model and conduct experiments to evaluate the effectiveness of our reward-based dialogue system for negotiation. Our results demonstrate that the proposed approach and reward system significantly enhance the agent's negotiation capabilities. The INA successfully engages in integrative negotiations, displaying the ability to dynamically adjust prices and negotiate the inclusion or exclusion of items in a bundle deal[1].

## 1 Introduction

In an online marketplace, customers and sellers engage in discussions involving product inquiry and bargaining before reaching a common consensus

(He et al., 2018). In such a setting, negotiation between the customer and the seller is a core facet of discourse that ultimately decides the profit of sale and customer satisfaction. Negotiation on the price of a product is very common, however, customers have an open-ended approach to negotiation often also involving negotiation on certain aspects related to the deal. For example, while buying a chair the customer may negotiate a deal without the cushions, or even negotiate between delivery and in-store pick-up. As a result, a dialogue system for negotiation in an online marketplace should be capable of engaging in negotiation on different aspects such as price, product, and delivery. Additionally, such a system should also be capable of responding to product inquiries with relevant and knowledge-grounded information.

A systemic survey conducted by (Zhan et al., 2022) discussed various datasets, evaluation metrics, and methodologies in common literature. From this, it can be implied that bargain in the marketplace typically follows a "Distributive" strategy where each party involved aims to maximize their gain rather than mutually benefiting outcomes. This strategy follows a win-lose model, where one party can gain only if the other party loses. The CraigslistBargains dataset (He et al., 2018) is the most prominent dataset in the price bargain domain with other datasets having less than 1,000 dialogues. This dataset contains dialogues between two human agents assigned the role of customer and seller negotiating over a product on Craigslist, the strategy used in the dialogues are largely distributive in nature. In contrast to a distributive approach, an "Integrative" approach to negotiation aims to reach a win-win situation by understanding the other party's needs and reaching a mutually satisfying consensus. It has been shown that an integrative approach to negotiation in retail e-commerce is more effective and leads to better customer satisfaction, than distributive approaches (Guttman

---

[1]Codes and dataset available at https://github.com/zishan-ai/neg and https://www.iitp.ac.in/~ai-nlp-ml/resources.html#INA

and Maes, 1998) that typically utilize agents that negotiate only on price. It is common in online marketplaces for products to have several items, such as a *"A chair and its cushion"*, a negotiation agent that is capable of satisfying customers that only want select items from the product such as customers that only want a chair or customers that only want a cushion is beneficial since the agent better understands customer requirements and may lead to win-win outcomes. Hence, treating a product as a "bundle" of items that customers can choose is a more integrative approach than treating the product as a single entity.

To incorporate this integrative approach, in this paper, we propose a novel dialogue system for negotiation in the online marketplace domain, which can respond to customers' inquiries and engage in negotiation with the customer. Unlike existing systems (He et al., 2018) that primarily focus on negotiation over the price of a product, our system follows a more integrative approach wherein negotiation involves different aspects such as adding or removing products from the aforementioned "bundle" of products, the price of the bundle, and the delivery of the product. Datasets for negotiation such as the CraigslistBargains dataset do not explicitly model the product as a bundle of smaller items. Hence, we construct a dataset (**IND**) consisting of integrative negotiation dialogues where the deal is modeled as a bundle of products. To avoid complete manual data creation, we design prompts for the GPT-J model (Wang and Komatsuzaki, 2021) to generate integrative negotiation utterances. To ensure the dataset's quality, we use humans in the loop for minor edits and filtering of the generated dialogues.

Using the constructed dataset, we build an integrative negotiation-powered dialogue agent (**INA**) using a supervised learning (SL) + reinforcement learning (RL) approach. To train our system, we leverage a novel reward function and maximize it using PPO loss (Schulman et al., 2017) to ensure aspects of negotiation consistency, negotiation power, and intent consistency. As per our knowledge, this is the first attempt to build an integrative-negotiation based dialogue system. Therefore we present a pioneering effort in developing an integrative-negotiation-based dialogue system, making several key contributions. *First,* we introduce a new task of integrative negotiation, expanding the scope of dialogue system research. *Second,* we propose an efficient approach for automatically generating data with minimal manual intervention, addressing the challenge of data scarcity in certain domains. This contribution will drive the development of more robust dialogue systems. *Third,* we create a unique dataset of integrative negotiation dialogues. *Finally,* we leverage the strengths of both supervised and reinforcement learning to construct a powerful dialogue system empowered by integrative negotiation strategies.

## 2  Related Work

Thompson et al. (2010) studied the effects of various intra-personal processes, such as mood, and interpersonal processes, such as emotion, on negotiation outcomes. They defined integrative negotiation as *"the extent to which the negotiated outcome satisfies the interests of both parties in a way that the outcome cannot be improved upon without hurting one or more of the parties involved"*. They also reported that the studies on the effectiveness of computer-mediated negotiation with respect to face-to-face negotiation give mixed results. Laroche and Genevay (2016) stated the importance of user adaptation in negotiation dialogue systems by performing experiments using different policies on simulated users in a newly designed negotiation dialogue game. Zhao et al. (2018) proposes a semi-automatic negotiation wherein a dialogue manager decides the intent after which a natural language generator presents conversational strategies to a human expert that writes the final utterance. Lewis et al. (2017) prepares a dataset and proposes end-to-end dialogue systems for "multi-issue bargaining". In this type of bargaining, two agents are presented with a set of items and asked to assign each item to one agent, each agent is also given a value function to decide the value of an item. He et al. (2018) prepares the CraiglistBargains dataset where two human agents negotiate over the price of a product listed on Craigslist, further, they decouple negotiation strategy and dialogue generation by proposing a dialogue manager to decide the intent of the next utterance and a generator that uses the intent to generate the utterance. Following this work, Yang et al. (2020) proposes a framework to integrate *"Theory of mind"* (Premack and Woodruff, 1978) for inferring personality types to enhance negotiation dialogues.

Unlike these previous works, our proposed negotiation agent (INA) is capable of doing integrative

negotiation. Our agent is not only capable of negotiation with respect to the price of an item but can also modify the deal to better suit the customer's preference. Similarly, our agent can also handle the customization of a deal proposed by the customer and decide on accepting or rejecting the deal. These capabilities are currently absent in any negotiation agent.

## 3 Dataset Creation

We construct the **IND** dataset for the task of integrative negotiation. To save on human effort and resources , we come up with a novel mechanism based on prompting a large language model for dataset creation. We keep human annotators in the loop only for making minor edits and filtering the automatically generated dialogues to ensure the quality of the conversations. The overall process consists of creating a skeleton of dialogues by dynamically deciding the correct intent for any arbitrary conversation. Our overall dataset creation process consists of 5 steps: (i). Background Data Creation, (ii). Intent Definition, (iii). Dialogue Flow Generation, (iv). Prompting for Dialogue Generation, and (v). Data Correction.

### 3.1 Background Data Creation

Although our method can be adapted to any product negotiation, we mainly focus on a list of 10 different electronic items: (i). Air Conditioning, (ii). Television, (iii). Refrigerator, (iv). Oven, (v). Washing Machine, (vi). Printer, (vii). Smart Phone, (viii). Laptop, (ix). Tablet, and (x). Camera. Along with these products, the deal bundle consists of a set of accessories related to the product. Therefore, our background database consists of the following information, such as Product Name, Product Description, Product Features, Price, Accessory List, and Accessory Description.

### 3.2 Intent Definition

In order to build a robust negotiation system it is vital to define intents that can cover a diverse range of scenarios during negotiation. For an integrative negotiation agent, the scenario in the scope of the agent is not just price negotiation, but also item-level negotiation in the given bundle. To cover these properties, we come up with the following intents[2]:

---

[2]Example utterances for each intent provided in Table 6 of the appendix.

- **Greet:** The utterances with general greetings like welcome and thank you come under this category.

- **Ask:** This intent is triggered when a user explicitly asks for information about an item or the ongoing negotiation.

- **Inform:** The agent may use the 'inform' intent to share detailed information about the products or services involved in the negotiation.

- **Ask-Clarification:** This intent captures the user's intention to seek further explanation or clarification regarding certain aspects of the negotiation or the overall deal according to the current negotiation state.

- **Negotiate-Price-Increase:** This intent indicates that the agent is seeking to increase the pricing terms of a product or service during the negotiation process.

- **Negotiate-Price-Decrease:** This intent indicates that the agent is seeking to decrease the pricing terms of a product or service during the negotiation process.

- **Negotiate-Price-NoChange:** This is an intent by the agent in a negotiation system indicating the system's intention to propose or assert that the price of a product or service should remain unchanged during the negotiation process. This is ideally done by highlighting the value and fairness of the current deal.

- **Negotiate-Add-X:** This intent by the agent or user refers to the intention to propose or suggest the addition of a specific item or feature to enhance the value of a product or service during the negotiation process. This may or may not lead to an increase in the price of the deal.

- **Negotiate-Remove-X:** This intent by the agent or user in refers to the intention to propose or suggest the removal of a specific item or feature from the deal in the negotiation process. This may or may not lead to a decrease in the price of the deal.

- **Accept:** This refers to the agent or user's intention to agree or accept a proposal, offer, or condition reached during the negotiation process.

**Product bundle:** [A tablet called Lenovo Tab P11 Pro. 11.5-inch OLED display, Snapdragon 730G processor ..., A Stylus pen called Adonit Note+ . Compatible with iOS and Android devices. Has 2048 levels of pressure ..., A Memory cards called Lexar Professional 633x SDXC UHS-I Card . This memory card has a capacity of up to 1TB ...]
**Initial price:** 92800
**Final price:** 83300

**User:** Hello, I'm interested in purchasing your Lenovo Tab P11 Pro. How much can I get it for? *Intent: Greet-Ask*

**Bot:** Hello! The Lenovo Tab P11 Pro along with a stylus pen and memory card is being sold for $92800.0 *Intent: Greet-Inform*

**User:** Excuse me, I do not need the stylus pen, is it possible to buy the tablet without the stylus pen? What would be the price? *Intent: Negotiate-Remove-X*

**Bot:** That is a very popular product, I can sell you the tablet without the stylus pen for $91100.0. *Intent: Inform*

**User:** That is a very expensive tablet, I'm afraid I can't afford it. How about $74700? *Intent: Negotiate-Price-Decrease*

**Bot:** I understand that it is a very expensive tablet. However, I'm willing to give you a great deal at $83300. Please let me know if you would like to purchase it. *Intent: Negotiate-Price-Increase*

**User:** This is perfect for me. I would like to buy it. *Intent: Accept*

**Bot:** That's great, we can proceed with the payment then. *Intent: Acknowledge*

Figure 1: An example conversation between the negotiation agent and a customer

- **Reject:** This refers to the agent or user's intention to agree or reject a proposal, offer, or condition reached during the negotiation process.

The above intents can occur either individually or in combination with other intents (e.g.: Greet-Ask).

## 3.3 Dialogue Flow Generation

Our dialogue flow generator module assumes that the dialogue flow (intent sequence) during negotiation can be random. However, we also put some obvious constraints on this dataset-generation process. One simple constraint is that the conversation would be initiated by the customer with a greet intent. This greet intent could be accompanied by a request for clarification or one of the 'negotiate' intents for the customer. The agent can respond by the inform intent or one of the agent 'negotiate' intents.

For all the deal bundles, we maintain negotiation details of the ongoing deal with the customer, which consist of: (i). Minimum Seller price, (ii). Current Seller price, (iv). Tolerance value ($tol$) and (iii). Current Customer price. To enforce the integrative nature of our agent, we limit only price-based negotiations to $d$ turns after which the 'Negotiate-Add-X' or 'Negotiate-Remove-X' intents would take over. To propose a price for the next turn, we assume that a decay in price difference (increment for customer and decrement for seller) over dialogue turns. This is in line with Faratin et al. (1998) where a similar function is used to model the price negotiation between the customer and seller. Equations 1 and 2 are used

for the computation of the proposed price by customer ($P_b$) or seller ($P_s$) at dialogue turn $t$. In the equations, $k$ is a constant to control the rate of price change from one turn to the next. If it $k$ is larger there will higher rate of concession, at a low value the rate of concession provided by the seller is low. For our setting, we have assumed a higher $k$ value for the seller and a lower $k$ for the customer, considering the customer is strict with their budget.

$$Ps_t = Pb_{t-1} + (Ps_{t-1} - Pb_{t-1})e^{-kt} \quad (1)$$

$$Pb_t = Ps_{t-1} - (Ps_{t-1} - Pb_{t-1})e^{-kt} \quad (2)$$

The seller will choose intent 'Accept' when the customer offered price is less than or equal to the amount $Ps_t - tol * Ps_t$. The customer will choose intent 'Reject' when the conversation has crossed the negotiation deadline, and the seller is no more ready to lower the bundle price. The dialogue flow terminates with the acknowledgment of 'accept' intent or the 'reject' intent.

## 3.4 Prompting for Dialogue Generation

We design few-shot prompts (Brown et al., 2020)[3] for each intent, with around four shots for each prompt (due to the token limit of 2,048 in GPT-J). Each shot contains three parts, a description of the task, a summary of the relevant information from the dialogue, and an utterance following the intent, all in a natural language format. The summary of the relevant information is designed considering the intent flow of the previous utterances of the dialogue. The description of the task is the sentences

---

[3]Example prompts provided in Section B.1 of the Appendix

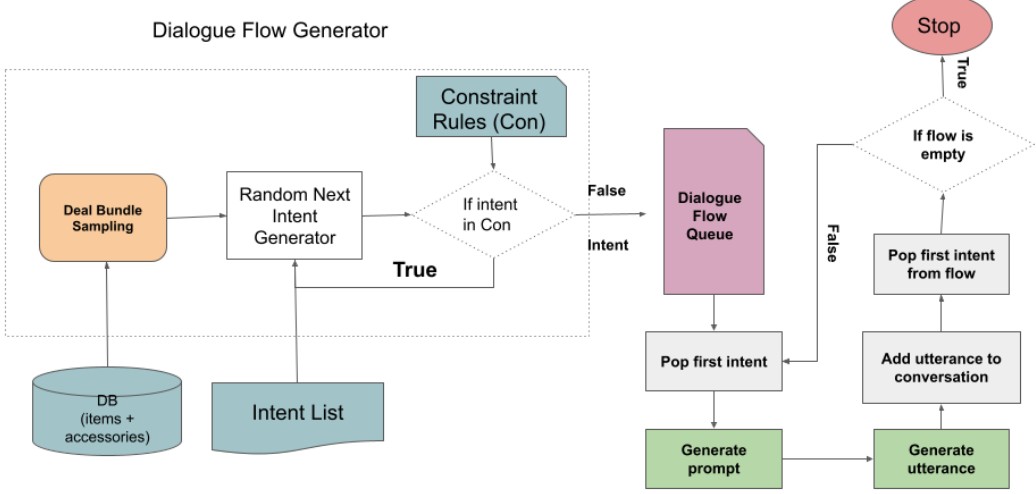

Figure 2: Overall data creation process

in the prompt that explains the situation and the goal of the intent, for instance, the task description for the *"Acknowledge acceptance"* intent is *"A customer has agreed to purchase a product from a seller, the seller wants to thank the customer and proceed with the transaction"*. The utterance following the intent is a manually designed utterance following the task description and the information summary of the shot.

The flow generation module creates an ordered list of intents along with relevant information for each intent, for instance, for the intent *"Negotiate-Add-X"* the item to be added is mentioned, and for *"Negotiate-Price-Decrease"* the price to be proposed is mentioned. Our algorithm uses the list created by the flow generation module to create a shot that is augmented to the prompt of the respective intent, this prompt is then passed to the GPT-J model to produce the utterance.

### 3.5 Data Correction

To ensure the quality of the automatically generated dataset, we implemented manual correction and filtration steps. We engaged three human experts who possess post-graduate qualifications and have two years of experience in the field. Their instructions were to make edits to the generated dialogues in order to ensure grounding in the provided background database, intent, action, and negotiation flow. Additionally, any utterances produced by the agent that referred to its own experiences or feel-ings, pretending to be human, were to be rephrased or removed (to maintain authenticity). The experts were also responsible for correcting minor grammatical errors. Furthermore, they were asked to rate the fluency of each utterance on a scale of 0-2, where 0 represented non-fluency and 2 indicated complete fluency. Dialogues containing utterances rated as 0 fluency were dropped from the dataset. These measures were implemented to uphold the quality standards of the dataset.

## 4   Dataset Statistic

The statistics of the dataset created are given in Table 1. The dataset has a total of 4,163 utterances and we follow an 80:12:8 split between train, test, and validation sets. The average number of turns per dialogue in the dataset is 13 and the number of unique words in the dataset, excluding numbers is 12,219, both these metrics are comparable to the metrics in the Craigslist Bargain dataset (avg. turns:9; unique words:11,799). Following (Wang et al., 2021), to automatically measure the **variability** of conversations of our dataset, we compute BLEU-1 and METEOR scores between the utterances. We obtain low BLEU-1 and METEOR scores of 0.08 and 0.05, respectively, indicating high variability between the utterances in IND. We ask three human experts to rate the **'engagingness'** and **'fairness'** of dialogues on a scale of 1 to 3 (higher the better). The dialogues obtained an average rating of 2.17 for 'engagingness' and 2.26 for

| | Train | Test | Valid |
|---|---|---|---|
| #Dialogues | 3330 | 500 | 333 |
| #Utterances | 45,914 | 6887 | 4592 |
| Avg # of words in Customer Utterance | 19.30 | 19.32 | 19.29 |
| Avg # of words in Sales-Person Utterance | 33.13 | 33.32 | 33.27 |

Table 1: Statistics of the dataset created (IND) 'fairness'[4].

## 5 Methodology

To force a language model to negotiate with the user while following its own price goal as well as approach, we fine-tune it using a novel-designed reward function in a reinforcement learning setting. Here, first, a pre-trained language model (GPT-2-medium) is fine-tuned in a supervised setting using traditional cross-entropy loss between the ground truth and predicted utterances probability distributions. For a supervised dialogue dataset $D = \{d_0, d_1, .., d_N\}$, where, $d = \{a_0, u_0, .., a_i, u_i, .., a_{T-1}, u_{T-1}\}$ - a multi-turn dialogue with $u_i + cxt_i$ ($u_i$ - user's utterance at $i^{th}$ turn and $cxt_i = \{a_0, u_0, .., a_{i-1}\}$) as input and $a_i$ (agent's utterance at $i^{th}$ turn) as output. The supervised learning dialogue model $\rho_\theta(d)$ can be expressed as:

$$\rho_\theta(d) = \prod_{T=0}^{T-1} \rho_u(u_i|u_{<i}, a_{<i})\rho_a(u_i|u_{<=i}, a_{<i}) \quad (3)$$

where $\rho_u$ and $\rho_a$ are the user's and agent's utterances probability distributions. This trained SLDM is fine-tuned in an RL setting using the PPO loss formulated as below:

$$L^{\text{CLIP}}(\theta) = \hat{E}[\min(pr_r(\theta)\hat{A}_r, \text{clip}(pr_y(\theta), \\ 1 - \varepsilon, 1 + \varepsilon)\hat{A}_r)] \quad (4)$$

where $pr_r(\theta) = \mathcal{P}_\theta^{new}/\mathcal{P}_\theta^{old}$. $\varepsilon$ and $\hat{A}_y$ denote the clipping range and normalized rewards, respectively. Finally, parameters' updation is done as follows:

$$\theta_{k+1} = \underset{\theta}{\arg\max} \underset{s,a\sim\mathcal{P}_{\theta_k}}{E}[L^{\text{CLIP}}] \quad (5)$$

Here, normalized rewards are obtained by a novel designed reward function ($R$) incorporating intent consistency reward ($R_1$), price gap reward

---

[4]The overall inter-annotator agreement using Krippendorff's alpha (Krippendorff, 2011) was found to be 0.84

---

($R_2$), negotiation strategy reward ($R_3$) and interactiveness ($R_4$) in generated responses. $R$ intuitively nudges SLDM towards these aspects by providing appropriate respective aspect penalization/reward for generated responses. For example, if the model generates intent inconsistent response then $R_3$ will penalize the model to discourage it from generating a similar type of content. All five rewards can be written as:

**Intent consistency:** In a negotiation system with complex intents there can often be divergence between the predicted intent and the intent of the generated utterance. To enforce this consistency, we propose Intent Consistency (IC) reward. This reward function is implemented by first training a BERT model (Devlin et al., 2018) on the training set of IND for the task of intent prediction. This task is modeled as a classification task where the input to the BERT model is an agent utterance at turn $t$, $Ua_t$, and the expected output is the intent of the utterance $Ia_t$. The accuracy of the trained intent classifier is 71.2%. We use the $[CLS]$ token for computing the probability distribution of the intent classes. We sample the probability value $P_{it}$ of the intent predicted $i$ by our end-to-end SLDM dialogue model and use it as $R_1$ (Eq. 6).

$$R_1 = P_{it}(u_t) \quad (6)$$

**Price Gap Reward:** The purpose of negotiation is to find a win-win solution for both the customer and the seller. The winning scenario for a seller would be as little reduction in the initially proposed price as possible. In line with this logic, we propose a Price Gap (PG) reward. This reward is simply the fraction of the initial proposed price by the agent $P_{ai}$ and the final selling price after negotiation $P_{af}$ (Eq 7). The higher the final price the greater the reward.

$$R_2 = \frac{P_{af}}{P_{ai}} \quad (7)$$

**Negotiation Strategy Reward:** A successful negotiation might not always entail deal acceptance. In cases where the customer wants to go below the minimum selling price of the agent $P_{a-min}$ it would not be judicious for the seller to satisfy the customer. In such situations where the negotiation could result in a win-lose situation, the deal should be rejected. Hence, the success criterion of the negotiation lies in not just acceptance of the deal but also the fairness of the deal. To ensure that our negotiation succeeds only in win-win scenarios we design the Negotiation Strategy (NS) reward.

$$R_3 = F(\frac{P_b - P_{a-min}}{P_{a-min}})G(Intent_f) \quad (8)$$

$$G(Intent_f) = \begin{cases} 1, & Intent_f = accept \\ -1, & Intent_f = reject \end{cases} \quad (9)$$

$$F(x) = \begin{cases} 0, & x < 0 \\ e^x, & x \geq 0 \end{cases} \quad (10)$$

In the above equations, $P_b$ is the customer's proposed price, and $Intent_f \in \{Accept, Reject\}$ is the final intent in the conversation used to capture the negotiation result. The reward incentivizes acceptance of a deal when the negotiated price is within the limit of a minimum price for the seller, and rejection when the negotiated price is below this minimum price.

**Interactiveness:** To ensure interactiveness, repetitions, and conversation blackholes are penalized such that system can engage the user for a longer duration with interactive responses. To penalize the generation of similar utterances for a given intent in the dialogue we use Equation 11.

$$R_4 = 1 - \frac{\sum_{i=1}^{i=m} \frac{v_k^{in} \cdot v_i^{in}}{|v_k^{in}||v_i^{in}|}}{m} \quad (11)$$

where $v_k^{in}$ is the vector (bag of words) representing the generated utterance with intent $in$. $v_i^{in}$ to $v_m^{in}$ are the vectors representing the previously generated utterances in the dialogue with the same intent. The final normalized reward function $R$ is formulated as:

$$R = \gamma_1 R_1 + \gamma_2 R_2 + \gamma_3 R_3 + \gamma_4 R_4 \quad (12)$$

with $\gamma_1 + \gamma_2 + \gamma_3 + \gamma_4 = 1$.

# 6 Experiments

## 6.1 Evaluation Metrics

To properly assess INA's performance, we perform both automatic and manual evaluations. In automatic evaluation to measure the surface similarity with the gold responses, we compute **METEOR** (Banerjee and Lavie, 2005). For semantic similarity, we compute **BERT Score (BS-F1)** (Zhang et al., 2019) and **Word Mover distance (WM)**. We also report the **Perplexity (PPL)** and the **average response length (R-LEN)** of the generated responses.

Human evaluations were conducted by three postgraduate evaluators who possess proficiency in similar tasks. Each evaluator interacted with the proposed system 15 times and assessed the conversations based on: **(i). Negotiation Consistency (N-Con):** It is the measure of consistency (absence of arbitrariness) in the negotiation approach within a dialogue **(ii). Bargaining Efficacy (B-Eff):** It measures the ability of the negotiation system to present compelling arguments, reasoning, or incentives that influence the other party's decision-making process., **(iii). Outcome fairness (O-fair):** It assesses the fairness or equity of the final outcomes reached during the negotiation process., **(iv). Dialogue-fluency (D-F):** It measures the overall grammatical correctness of the generated responses, and **(v). Dialogue-Engagingness (D-E):** Measures the extent to which a conversation or dialogue is interesting, captivating, and able to hold the attention of the participants. The evaluators assigned scores on a scale of 1 to 3 for each metric (The higher the better).

# 7 Results and Analysis

**Automatic Evaluation:** It can be noticed from Table 2 that the proposed **INA** performs better than all the four baselines *viz.* ARDM, ARDM + BK (Background Knowledge), ARDM + In (Intent), and Neg-TOD (Hosseini-Asl et al., 2020), in terms of all the five metrics *viz.* **METEOR**, **BS-F1**, **WM**, **PPL**, and **R-LEN**. For the evaluation metrics measuring similarity (of the generated utterance) with the gold utterance *i.e* METEOR, BS-F1 and WM, **INA** attains scores of *0.43*, *0.86* and *0.57*, respectively. The obtained scores are significant improvements <0.141, 0.042, 0.04>, <0.158, 0.032, 0.04>, <0.144, 0.032, 0.03> and <0.137, 0.029, 0.03> over the baselines, ARDM, ARDM+BK, ARDM+In, and NegTOD, respectively.

It can also be inferred that the difference of **BS-F1**, and **WM** scores decrease in the following order: INA>INA-NS>INA-PG>INA-I. This shows the importance of task-specific rewards in our proposed system **INA**.

It can also be observed from Table 2 that **INA** obtains lower (better) **PPL** = 1.56 score than that of ARDM, ARDM+BK, ARDM+In, and NegTod with a difference of 1.39, 1.19, 1.24, and 1.37 points, respectively. Further, we obtain a score of **R-LEN** = 39.93 is also better than that of ARDM, ARDM+BK, ARDM+In, and Neg-TOD with a dif-

| Model | METEOR | BS-F1 | WM | PPL | R-LEN |
|---|---|---|---|---|---|
| ARDM (Wu et al., 2021) | 0.289 | 0.823 | 0.53 | 2.95 | 25.22 |
| ARDM + BK | 0.272 | 0.833 | 0.53 | 2.75 | 35.29 |
| ARDM + In | 0.286 | 0.833 | 0.54 | 2.80 | 30.30 |
| NegTOD | 0.293 | 0.836 | 0.54 | 2.63 | 32.32 |
| INA - IC | 0.39 | 0.833 | 0.54 | 2.16 | 39.15 |
| INA - PG | 0.31 | 0.832 | 0.53 | 2.34 | 39.11 |
| INA - NS | 0.33 | 0.833 | 0.54 | 2.05 | 39.08 |
| INA - I | 0.34 | 0.831 | 0.53 | 2.30 | 39.13 |
| **INA** | **0.43** | **0.865** | **0.57** | **1.56** | **39.93** |

Table 2: Results of automatic evaluation

| Model | N-Con | B-Eff | O-fair | D-F | D-E |
|---|---|---|---|---|---|
| ARDM (Wu et al., 2021) | 0.5 | 0 | 1.2 | 1.8 | 2 |
| ARDM + BK | 1.2 | 0.4 | 1.4 | 2.4 | 2.2 |
| ARDM + In | 0.8 | 0.4 | 1.2 | 2 | 2 |
| INA - IC | 1.8 | 1.6 | 1.4 | 2.7 | 2.4 |
| INA - PG | 2 | 1.2 | 1.4 | 2.6 | 2.4 |
| INA - NS | 1.8 | 1.6 | 1.6 | 2.7 | 2.1 |
| INA - I | 2 | 1.6 | 1.6 | 2.5 | 2.0 |
| **INA** | **2.4** | **1.8** | **1.8** | **2.8** | **2.6** |

Table 3: Results of human evaluation

ference of 15.72, 2.28, 13.5, and 1.76, respectively. This indicates that the **INA** is able to generate longer responses, hence, showcasing more engagingness with the user. It can be due to the incorporation of all four rewards where $R_1$, $R_2$, and $R_3$ play the crucial role in handling negotiation and price consistency, and $R_4$ helps in maintaining the non-repetitiveness, hence, driving the agent to build the rapport with a user as well as be on the goal by generating diverse and interactive negotiation responses.

**Human Evaluation:** Table 3 shows the human evaluation results for all the eight models *viz.* ARDM, ARDM+BK, ARDM+In, NegTOD, INA-IC, INA-PG, INA-NS and INA-I. It may be noted that **INA** yields better scores for **N-Con**, **B-Eff**, **O-fair**, **D-F**, and **D-E** compared to the baselines. Scores of **N-Con**: 2.4, **D-F**: 2.8, and **D-E**: 2.6 shows that the intent-consistency (IC) and interactiveness (I) rewards play a crucial role in obtaining consistent, fluent, and engaging, responses as compared to other models. Further, in terms of **B-Eff** and **O-fair**, INA attains a score of 1.8 for both. The ablation of the price-gap (PG) and negotiation-strategy (NS) rewards showcases the importance of these rewards in terms of B-Eff and O-fair. Therefore, it can be inferred that employing intent consistency, price gap, and negotiation strategy rewards help in a more consistent, persuasive, and overall fair negotiation with the customer.

## 8 Conclusion

In this paper, we have presented a novel dialogue agent for negotiation (INA) in the online marketplace domain, focusing on an integrative approach that goes beyond price negotiations. Our system can respond to customer inquiries and engage in negotiations that encompass various aspects, such as modifying product bundles, adjusting prices, and arranging product delivery. Unlike existing systems that mainly concentrate on price negotiations, our approach provides a more comprehensive and versatile solution. To achieve this, we constructed a dataset of negotiation dialogues (IND) where the product is represented as a bundle of smaller items. To minimize manual effort in data creation, we employed prompts for the GPT-J model to generate integrative negotiation utterances. Using IND, we developed a INA through a combination of supervised learning and reinforcement learning. Our training process incorporated a novel reward function that suits the negotiation task, which we optimized using the Proximal Policy Optimization (PPO) loss. Our results show that INA is able to perform integrative negotiations with the customer and enable engaging negotiations that can lead to a win-win deal for the seller and the customer.

In the future, it would be interesting to explore the role of the customer persona like age, gender, hobbies, etc. during negotiation.

## 9 Limitations

Our data creation steps and modeling have some limitations. First, to create the data, GPT-J is used which requires a large GPU memory size (here, 40 GB). Another limitation of GPT-J is that it has a context window of 2,048 tokens, which constrains our prompting mechanism. Within this context window, we need to fit background data as well as dialogue history with a few shot examples. This allows us to only go for a maximum of 4 shots while prompting leading to some hallucinations in the created data which needed to be fixed manually.

## 10 Ethical Considerations

Since negotiation by nature entails bargain with the customer, it should be done ethically. Our integrative approach to negotiation gives greater flexibility to the customer and hence leads to a win-win scenario in negotiation. Our negotiation is not aimed at as a zero-sum game where a party has to lose in order for the other to win. The customer at any point of the conversation can reject the deal and thus is not compelled to continue with the negotiation if it does not suit them.

The dataset created in this work will be made available only after filling and signing an agreement declaring that the data will be used only for research purposes. The annotation, filtering/editing of data, and manual evaluations were done by human experts, who are regular employees of our research group and are paid in accordance with the institute's policy. There are no other issues to declare.

## 11 Acknowledgement

Authors acknowledge the grant received from Accenture LLP for the project T"Conversational Agents with Negotiation and Influencing ability".

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

# A Appendix

## A.1 Implementation Details

For generating the **INA** corpus, GPT-J model (Wang and Komatsuzaki, 2021) with 6 billion parameters was used. **INA** is trained in an RL framework by employing a fine-tuned GPT-2 small (Radford et al., 2019) model (117 million parameters) on our proposed **IND** dataset. For dialogue flow generation, we keep the value of $d$ as 2. In each iteration of RL-training, $n = 3$ candidate responses are generated. It is selected as per PPL score, after experimenting with different values i.e. $n = 2, 3, 4, 5, 10$. Nucleus sampling (Holtzman et al., 2019) with temperature $T = 0.8$ and probability $p = 0.9$ is used to decode the generated utterances. **INA** trained is done using $seed\_value = 10$, $human\_reward = 10$, $max\_candidate\_length = 50$ with $optimizer = AdamW$ (Loshchilov and Hutter, 2017) and learning rate $\alpha = 2e - 05$, $\varepsilon = 0.2$ and $epochs = 17$. The reward weight combination of $0.2, 0.2, 0.3, 0.2$ are chosen as the final weights for $\gamma_1$, $\gamma_2$, $\gamma_3$, and $\gamma_4$ respectively.

### A.1.1 Specifications of Computational Resource

To train the MLE-loss-based conversational model, and proposed **INA**, following configurations are used:

- **GPU:** A100-PCIE-40GB.

- **CUDA Support:** CUDA 11.x (or later.

- **Memory clock:** 1215 MHz.

- **Total board power:** 250 W.

- **GPU clocks:** Base: 765 MHz, Boost: 1410 MHz.

- **Memory Size:** 40 GB.

- **Memory Type:** HBM2.

- **Bus Width:** 5120 bits.

## B Dataset

We ensure that the utterances in INA are grounded on the background knowledge consisting of product and deal details. Table 5 shows example utterances from our dataset for different agent intents. It can clearly be observed that the utterances are well-grounded in the background knowledge and do contain factual hallucinations.

Table 6 shows example utterances for each intent defined in Section 3. The table also shows examples of custom intents obtained through the combination of the defined intents.

### B.1 Prompts

Each prompt contains around 4 shots and each shot contains a task description, required context, and an example utterance.

An example shot for the intent "Negotiate-Price-NoChange" is as follows:

"A seller is negotiating with a customer for a laptop called Dell X8, it has 16 GB ram, 11-inch screen and Intel i7 processor. The seller ideally wants it for $770 and is not willing to reduce the price."

"'The customer tells "I appreciate that you need to make a profit on this item, but unfortunately, $770 is above my budget for a laptop. I was ideally hoping to purchase the Dell X8 for $500, but I'm willing to negotiate up to $570 if necessary. Is there any way you could lower the price to meet me somewhere in the middle?"'"

| $k_{customer}$ | $k_{seller}$ | $utility_{customer}$ | $utility_{agent}$ |
|:---:|:---:|:---:|:---:|
| 0.2 | 0.8 | 0.73 | 0.45 |
| 0.4 | 0.6 | 0.64 | 0.53 |
| 0.6 | 0.4 | 0.51 | 0.65 |
| 0.8 | 0.2 | 0.33 | 0.82 |

Table 4: Table showing price simulation result for different value of k for counter offer generation for seller $k_{seller}$ and buyer $k_{buyer}$, the most practical and fair outcome where average utility (final price above reserve preference) was not one sided in more than 50% simulations we were getting was on value 0.4 for seller and 0.6 for buyer. One assumption we had taken was customer will be more stricter to his/her budget. Therefore customer will increase his/her budget in counter offer in lower rate than seller will decrease the price of bundle in counter offer. Final value of k was 0.4 for customer and 0.6 for seller.

"(Remember, the seller cannot go lower than $770) The seller endorses the product by saying."

"<start> While I appreciate your offer, I cannot go as low as $570. I cannot lower the price further since the laptop is high-end and is well worth $770. It has 16 GB ram and an Intel i7 processor, making it ideal for heavy duty applications. I'm sure that you would be pleased with it even for $770!"

Here, we provide context in terms of the previous utterance of the customer and the product description. The task mentioned is to endorse the product and not go below a particular price.

An example shot for the intent "Negotiate-Remove-X" is as follows:

"A customer is negotiating with a seller about a product. The customer wants to ask for another deal to the seller."

"The initial deal was a laptop called Dell X8, it has 16 GB ram, 11-inch screen and Intel i7 processor along with a gaming mouse. The price for this deal was $800. The customer wants to remove the gaming mouse from the deal."

"The customer asks for the new deal by saying."

"<start> I do not really need the mouse, is it possible to just sell me the laptop?"

Here the task description is mentioned explicitly in 3 lines. The context is provided as the previous deal.

## C   INA Generated Conversation Sample

In Table 7 we show one sample interaction between a human and INA. The negotiation is for a bundle deal of a TV, TV Stand, and Extension Cord. The interaction shows the capability of our dialogue agent (INA) in handling and integrative negotiation.

| | |
|---|---|
| **Background DB** | 1. An Oven called Frigidaire FGET3066UF. This double-wall oven has a total capacity of 10.2 cu. ft. and features a true convection cooking system with a third heating element and a fan. It also has a built-in temperature probe, a self-cleaning system, and a steam cleaning option. (Main product)

2. A Microwave Safe Coffee Maker Mug called Corelle Coordinates Microwave Safe Coffee Mug. This 11-ounce coffee mug is made of durable, lightweight stoneware and is microwave safe. It features a simple, classic design and is dishwasher safe for easy cleaning (Side product 1)

3. A Microwave oven cleaner called Weiman Microwave & Cooktop Cleaner. A multi-purpose cleaner that works on both the inside and outside of your microwave. The formula is designed to dissolve tough stains, grease, and grime, leaving your microwave looking like new. (Side product 2) |
| **Utterance** | I understand your budget constraints, but I'm sorry I cannot reduce the price of the oven. It has a total capacity of 10.2 cu. ft. and features a true convection cooking system with a third heating element and a fan. It also has a built-in temperature probe, a self-clean system, and a steam cleaning option. along with a microwave oven cleaner called Weiman Microwave & Cooktop Cleaner. It is designed to dissolve tough stains, grease, and grime, leaving your microwave looking like new. [Intent: Negotiate-Price-NoChange] |
| **Utterance** | The microwave oven cleaner is a multi-purpose cleaner that works on the inside and outside of your microwave. It works to dissolve tough stains, grease, and grime, leaving your microwave looking like new. (Reply to: What is the Microwave oven cleaner?) [Intent: Provide_Clarification/Inform] |
| **Background DB** | 1. A AC called Voltas 1.5 Ton 3 Star Split AC. Comes with a 3-star energy rating, a cooling capacity of 5100 W, and an anti-dust filter. (Main product)

2. A Smart thermostat called Honeywell Home T9. The Honeywell Home T9 smart thermostat features multi-room sensors that measure temperature and humidity in different areas of your home, helping to keep your home at the right temperature in every room. It can be controlled using a smartphone or tablet and features built-in Amazon Alexa voice control. (Side product) |
| **Utterance** | While I appreciate your offer, I cannot reduce the price of the AC since it has a 3-star energy rating and a cooling capacity of 5100 W. It also has a built-in anti-dust filter. I'm sure that it will be worth the investment even for $23900.0 [Intent: Negotiate-Price-NoChange] |
| **Utterance** | Yes, the cooling capacity is 5100 W. (Reply to: Is the Cooling capacity of the Voltas 1.5 Ton 3 Star Split AC 5100 W?) [Intent: Provide_Clarification/Inform] |

Table 5: Examples showing the intents expressed in generated utterances in the dataset while being grounded in the background database

| Intent | Example |
|---|---|
| Greet-Ask | Hello, I am interested in buying the refrigerator you have listed. How much can I get it for? |
| Greet-Inform | Hi! I'm selling the Speed Queen TR3000WN washing machine for $11000.0 |
| Negotiate-Remove-X _Negotiate-Price-Decrease | I really like the phone but I don't need the charging case, earphones, or charging case. Can you sell me the phone for $42200.0 |
| Negotiate-Price-Remove-X | Certainly! I can remove the refrigerator and covers from the deal and sell the refrigerator to you, however, the price is $19300.0 |
| Negotiate-Remove-delivery | Excuse me, I do not want the delivery. Can I have the TV and stand without delivery? What would be the new price? |
| Negotiate-Price-NoChange | I understand your budget constraints. However, I'm sorry I cannot reduce the price of the Nikon Z7 II. It is a top-notch camera, and is one of the best available in the market. It has a full-frame mirrorless sensor, which gives it a 493 focus points and a 45.7 megapixel resolution. It also has a weather-resistant flap and a comfortable shoulder strap. It also has a 4K video recording feature, which makes it ideal for shooting video. I'm sure that it will be worth the investment even for $51300.0 |
| Negotiate-Price-Increase | I understand that you're looking for a refrigerator with a lower price. However, I'm afraid that the price of this refrigerator is already very low. I'm willing to reduce the price to $16700.0 Are you interested? |
| Ask_Clarification-Y | What's the refresh rate of this TV? |
| Provide_Clarification-Y | The refresh rate of this TV is 60Hz. |
| Negotiate-Add-X | What will be the total price if I buy the laptop together with the keyboard and the USB Hub? |
| Tell_price | For the laptop, external hard drive, and mouse it would be $34900.0 |
| Negotiate-Price-Decrease | I'm interested in the Samsung QN90A Neo QLED TV. Unfortunately, I'm not willing to go above $78600.0 Is there any way you could meet me in the middle and sell it to me at $68500.0 |
| Reject | Sorry, my budget is limited to $29800.0 I'm sure you will find better deals, thank you for your interest. |
| Ask_Price | What would be the price for the thermostat? |
| Negotiate-Remove-X | I do not really need the stylus, is it possible to just sell me the tablet? |
| Accept | I'm willing to purchase the laptop at that price. Let's go ahead with the transaction. |
| Acknowledge acceptance | Great! I am pleased that you are buying and I'm sure you will be happy with your purchase. |
| Greet-Ask _Negotiate-Price-Decrease | Hello, I'm interested in buying your Samsung Galaxy S21 Ultra 5G, but I was hoping to get a better price. Can I get it for $62900.0 |
| Avoid_rejection | Ok, in that case we can proceed with $39500.0 Shall we proceed to checkout? |
| Greet-Inform _Negotiate-Price-NoChange | Hello! I'm glad you're interested in the washing machine and dryer. While I understand your budget, I'm sorry I cannot lower the price. This is an amazing machine, which is why I can only offer $23200.0 While I understand that it's a bit above your budget, I'm sure my customers will love it. |
| Greet-Inform _Negotiate-Price-Increase | Hello there! Thank you for your interest in the A Carrier 1.5 Ton 3 Star Split AC. I'm sorry, but I am not able to go down to $53400.0 but I am willing to lower the price to $57700.0 If you are interested, please let me know. |

Table 6: Example utterances for each intent type

| | Product Description: A TV called Skyworth Q71 Series TV. This 4K TV features a 60Hz refresh rate, 4K resolution, 65-inch screen, and Dolby Vision and Atmos for enhanced audio and visual experiences., (A TV Stand called WALI Universal TV Stand. This stand fits most TVs from 22 to 65 inches and can support up to 110 pounds. It has a tempered glass base, adjustable height, and cable management system., An Extension Cord called AmazonBasics Indoor Extension Cord. This is a 6-foot extension cord that can handle up to 13 amps , buyer asking for: TV, (TV Stand, Extension Cord)

Bundle 1: TV + (TV Stand, Extension Cord)
Initial Buyer Proposed Price: 55000
Initial Seller Proposed Price: 67000
Minimum Seller Price: 57000

Bundle 2: TV + (TV Stand)

Initial Buyer Proposed Price: 52500
Initial Seller Proposed Price: 65000
Minimum Seller Price: 55000

Bundle 3: TV + (Extension Cord)
Initial Buyer Proposed Price: 53000
Initial Seller Proposed Price: 65500
Minimum Seller Price: 55500

Bundle 4: TV
Initial Buyer Proposed Price: 51500
Initial Seller Proposed Price: 645000
Minimum Seller Price: 53500 |
|---|---|
| Context | |
| Buyer | Hello, I'm interested in purchasing the Skyworth Q71 Series TV, Bose Soundbar 700, the GE Designer Extension Cord |
| Agent | Hello! I'm glad to hear that you're interested in the Skyworth Q71 Series TV, Bose Soundbar 700, the GE Designer Extension Cord |
| Buyer | I'd like to know about the TV. What is the resolution of the model? How much does it cost? |
| Agent | The TV has 4K resulution. The TV includes a remote control and a remote shutter release, will cost you $66600 |
| Buyer | I am interested in buying the set of TV, but I do not want the TV stand. What will be the cost? |
| Agent | I understand that $66600 is above your budget. However, I'm willing to meet you in the middle at $64,600 after removing TV stand |
| Buyer | Also I do not need the extension cord, can we remove extension cord from the deal.Also |
| Agent | Yes, we can remove extension cord, it will cost you $62000 |
| Buyer | I have a budget of $59,500. Can you give me more concession? |
| Agent | Certainly! I can offer you the TV from the deal for $61,000, however, $59,000 is too less |
| Buyer | I cannot afford more than $59500, You should understand I am interested in this deal, could you finalize the deal at $60000 |
| Agent | Ok, we can have a deal, Thank you |

Table 7: Sample conversation between a human and INA