# OpenReview forum: "INA: An Integrative Approach for Enhancing Negotiation Strategies with Reward-Based Dialogue Agent"
_EMNLP/2023/Conference — EMNLP 2023 Findings_

### Official Review · Reviewer_PVmB · 2023-08-02

**Typos Grammar Style And Presentation Improvements:** 1) 755
**Soundness:** 3

**Excitement:**

3: Ambivalent: It has merits (e.g., it reports state-of-the-art results, the idea is nice), but there are key weaknesses (e.g., it describes incremental work), and it can significantly benefit from another round of revision. However, I won't object to accepting it if my co-reviewers champion it.

**Paper Topic And Main Contributions:**

The authors propose a new dataset and dialogue agents for agent-human negotiations. The authors claim that their work looks at each product as a bundle of multiple items, which can all be negotiated together, unlike prior work that focuses on win-lose conversations over the price of products. The dataset is collected in a semi-automated way using randomly generated (with some sensible constraints) dialogue flows based on intents and subsequent utterance generation by prompting a language model. Their approach is based on a supervised + reinforcement learning paradigm using a combination of multiple relevant rewards. A comprehensive set of experiments is provided based on automated and human evaluation.

Overall, I believe this is an exciting direction to pursue, and the proposed dataset and approach are reasonable. However, the pitch of the paper is misleading, which questions the overall novelty and usefulness of the work.

**Questions For The Authors:**

1) How was k decided? What were the final values?

2) Eq 7 seems to define the reward at the end of the dialogue. How did the authors tackle the issue of back-propagation through such long token sequences? It is unclear how the RL rewards are defined currently. I suggest providing more details about the reward design for clarity.

3) The data collection makes several assumptions that hurt the diversity of the collected data, such as assuming concession rates for buyers and sellers. Further, semi-automatic methods make it challenging to capture other variables, such as user personality. I wonder if the authors have any thoughts on this and how it can be improved in the future - it would be nice to discuss this in the paper and can improve the overall contributions made in this work.

**Reasons To Accept:**

1) The semi-automatic data collection technique is an interesting step and potentially useful in this space where collecting quality data is challenging. This naturally has pros and cons (I discuss some of them below). The process followed was also rigorous, involving post-processing and human evaluation to ensure quality.

2) In their approach, the authors defined useful rewards that are all relevant to negotiation and can potentially help future work.

3) The paper consists of a comprehensive evaluation, showing the superiority of the proposed approach on a variety of metrics based on both automatic and human evaluation.

**Reasons To Reject:**

1) The proposed approach is unclear. RL formulation has not been stated properly. It is unclear what the state and actions are and whether the rewards are defined at an utterance level or a dialogue level. I suggest the authors to describe the approach in a more thorough manner.

2) The paper argues for win-win negotiations that “see” a product as a bundle rather than negotiating for a single issue of price. However, neither the proposed approach nor the evaluation focuses on the other items in the bundle. The rewards are based only on the price. In other words, the paper set out with the motivation of handling a scenario with multiple items in a bundle, but the spirit of the approach seems no different than prior work by He et al 2018. What am I missing here?

3) In addition, the pitch of the paper ignores a body of prior work on multi-issue negotiations that do capture multiple items rather than just one. These datasets already involve integrative negotiations (along with distributive negotiations). For instance, work on DealOrNoDeal task (Lewis et al. 2017) and more recent datasets like CaSiNo (Chawla et al. 2021) and JobInterview (Yamaguchi et al. 2021). These should be discussed in related work. One difference is that in this paper, the set of issues varies depending on the given context. Thus, the claims in Lines 126-128 and 189-191 are misleading.

To fix these, I believe either the pitch of the paper should be changed or the experiments should be better designed in a way that captures the multiple items in a bundle (just like modeling work on multi-issue negotiation tasks). One way to do this can be by defining a reward that captures the other items in the bundle.

4) The paper lacks additional critical analysis, such as comparing the intent accuracy of the final models and analyzing how the model deals with the other items in the bundle - this is important since this relates to how the paper has been motivated. Hence, I suggest the authors to include this in the paper.

**Reproducibility:**

3: Could reproduce the results with some difficulty. The settings of parameters are underspecified or subjectively determined; the training/evaluation data are not widely available.

**Reviewer Confidence:**

5: Positive that my evaluation is correct. I read the paper very carefully and I am very familiar with related work.

---

> ### Author Rebuttal · Authors · 2023-08-29
>
> 1)  In the proposed approach the rewards were defined at the dialogue level. We will endure to elaborate on the approach in a more thorough manner in the camera ready version.
> 2) In our proposed approach, negotiation is on the bundle, where a customer can add an item with some concession price or he/she can remove any item and go for the deal with primary item of the bundle. On adding any accessory to the bundle it is actually offering a reasonably negotiated price to the customer and also it is giving some better margin to seller (above the reserve preference of cost of bundle), that way our proposed approach is doing negotiation where one issue is definitely price but the other issue is addition and removal of an accessory from the bundle and negotiation on bundle with added accessory or removed accessory.
> 3) The test accuracy of BERT intent classifier is 71.2%, training:validation:test 2000:500:1500 instances, however we have not mentioned classification accuracy of proposed RL model but as per the human evaluation results, we can see on absence of intent consistency reward, model has shown below average bargaining efficacy and dialogue engagingness, which shown intent consistency contribute to the conversation or dialogue in makings it interesting, captivating, and able to hold the attention of the participants and also it helps in presenting compelling argument in response to the customer counter offer or issue with price. The proposed model is aiming for the best possible outcome for the customer and seller, therefore accessories addition and removal accessory decision is one of the factor which help in finalizing the deal, the decision is based on a counter offer from the customer. We will ensure to better explain this point in the camera ready version of the paper.
> 4) We have taken different value of k for counter offer generation for seller and buyer, We performed price simulation for different value of k for both seller and customer , the most practical and fair outcome where average utility (final price above reserve preference) was not one sided in more than 50% simulations we were getting was on value 0.4 for seller and 0.6 for buyer. One assumption we had taken was customer will be more stricter to his/her budget. Therefore customer will increase his/her budget in counter offer in lower rate than seller will decrease the price of bundle in counter offer. Final value of k was 0.4 for customer and 0.6 for seller. We will explain this process in detail in the camera ready version
> 5) Because of O(e^n) possibility of conversation during negotiation where a customer can add or remove accessory any number of items, we have taken one assumption of maximum 2 accessories in bundle however user have choice to add or remove 1 or 2 accessories. Final value of k is not purely assumption, we should have mention our basis of k value assumption for buyer and seller, since we have taken different value of k for counter offer generation for seller and buyer, we performed price simulation for different value of k for both seller and customer , the most practical and fair outcome where average utility (final price above reserve preference) was not one sided in more than 50% simulations we were getting was on value 0.4 for seller and 0.6 for customer. Our assumption here was customer is more strict to his budget then seller. However, concession can be given based on entailment of persona and the accessory if we have some prior information about the customer persona or we can capture the persona during conversation that is his/her likes and dislikes, based on that we can decide how much entailed offer we have for customer and eventually this will help in deciding the final concession on the counter offer from seller. In future work Artificial theory of Minds (AToM) will be also very helpful in deciding the counter offer for the customer.

---

### Official Review · Reviewer_gPQD · 2023-08-02

**Soundness:** 3

**Excitement:**

3: Ambivalent: It has merits (e.g., it reports state-of-the-art results, the idea is nice), but there are key weaknesses (e.g., it describes incremental work), and it can significantly benefit from another round of revision. However, I won't object to accepting it if my co-reviewers champion it.

**Paper Topic And Main Contributions:**

Towards negotiation in the online marketplace, this paper creates a new semi-automated data creation method and release a new dataset called Integrative Negotiation Dataset (IND). It further builds an integrative negotiation-powered dialogue agent (INA) with a process similar to RLHF.

**Reasons To Accept:**

This work provides substantial contents including:
1. A concrete pathway to collect data
2. A detailed training framework containing SFT and RL, as well as the delicate reward design.
3. Relatively positive experimental results both on automatic and human evaluation.

**Reasons To Reject:**

Figures in this paper should be polished up. For example, several lines in Figure 2 are actually crooked.

**Reproducibility:**

4: Could mostly reproduce the results, but there may be some variation because of sample variance or minor variations in their interpretation of the protocol or method.

**Reviewer Confidence:**

2: Willing to defend my evaluation, but it is fairly likely that I missed some details, didn't understand some central points, or can't be sure about the novelty of the work.

---

> ### Author Rebuttal · Authors · 2023-08-29
>
> 1) We thank the reviewer for pointing out the figures, we will make sure to fix this issue in the camera-ready version of the paper.

---

### Official Review · Reviewer_ndma · 2023-08-05

**Typos Grammar Style And Presentation Improvements:** Line 441
**Soundness:** 3

**Excitement:**

3: Ambivalent: It has merits (e.g., it reports state-of-the-art results, the idea is nice), but there are key weaknesses (e.g., it describes incremental work), and it can significantly benefit from another round of revision. However, I won't object to accepting it if my co-reviewers champion it.

**Paper Topic And Main Contributions:**

This paper introduces a new task of integrative negotiation along with an accompanying new dataset.  Most prior works establish agents that attempt to bargain for the best price possible during a discussion, resulting in a win-loss dynamic.  In contrast, this work designs an agent that attempts to complete a negotiation in a win-win manner, by bargaining on items on unequal value, such as partial items in a bundle.

The dataset is created by (a) coming up with 10 product categories, 11 turn-level intents, various pieces of meta-data such as item price (b) randomly select these intents to create a dialogue flow skeleton (c) use GPT-J to generate dialogues using 4-shot ICL following the dialogue flow (d) Authors finally review the generated data to ensure quality.   This results in ~4K total dialogues, with an average of 13 turns per dialogue.

For modeling, the authors first fine-tune a GPT-2 Medium on the ground truth dialogues.  This model is then further updated with PPO in an RL setting where the reward contains four parts.  (1) Intent consistency - train a BERT model to ensure that the intent for the dialogue flow is the one that is generated (2) Price gap - higher reward for higher price obtained.  (3) Negotiation strategy - nudges the model to only accept a deal when within a certain threshold (4) Interactiveness - a penalty (negative reward) for producing utterances that are too similar.

**Questions For The Authors:**

Any reason for choosing GPT-J over GODEL or Koala?

What is the accuracy of the BERT intent classifier?

**Reasons To Accept:**

Authors present a novel new dataset of reasonable size, containing discussion on integrative negotiation dialogues.  The dialogue contain clarification intents, which is frankly useful in itself as a totally different task worth studying.  3,300 training dialogues is a respectable amount.

For quantitative metrics (METEOR, BERT-Score, WMD, Perplexity, and Length), the proposed INA model outperforms all baselines.  In human evaluation across five qualitative metrics, the INA model also does better than all ARDM and NegTOD baselines.

**Reasons To Reject:**

The quality of the dialogues is still fairly unrealistic.  To highlight a couple of areas:
a) The users don't really dive deep into the description or product details.  (ie. "The SnapDragon processor is 2 generations old now, you should lower the price.")
b) There are no emotional appeals that would occur (ie. "Oh, please!  I really need it for my daughter's birthday gift!")
c) Conversations are all roughly the same length, real life chats can go on for 100+ turns easily (or conversely, will end abruptly).
d) The prices are off by a factor of 100x?  (ie. Tablet costs $92,800, rather than $928)

There are no ablations or extra analysis on the rewards used for RL-training.  In general, the modeling is not too novel, relying largely on pre-trained SL followed by RL, which is fairly common for dialogue system training.  This is not a huge deal though if we view this as resource paper, rather than a modeling paper.

**Reproducibility:**

3: Could reproduce the results with some difficulty. The settings of parameters are underspecified or subjectively determined; the training/evaluation data are not widely available.

**Reviewer Confidence:**

3: Pretty sure, but there's a chance I missed something. Although I have a good feel for this area in general, I did not carefully check the paper's details, e.g., the math, experimental design, or novelty.

---

> ### Author Rebuttal · Authors · 2023-08-29
>
> 1) Considering limited compute available in the research setting we could not use LLMs larger than 7 billion parameters. For this reason we could not use the Koala-13b model. The Koala-7b model (which we could use) is benchmarked lower than the GPT-6b model on the open-llm leaderboard. GODEL models on the other hand has a maximum size of 2.7 billion parameters and although trained for dialogue setting, our in context prompts consist of dialogue summaries instead of raw dialogues. For this reason GODEL was not found to be suitable for data generation purposes for the given task.
> 2) The test accuracy of BERT intent classifier is 71.2%, training:validation:test 2000:500:1500 instances. We will make sure to mention this in the camera ready version of the paper.

---

### Meta-Review · Area_Chair_pquh · 2023-09-20

**Recommendation:** 3

**Metareview:**

This paper contributes a valuable new dataset of negotiation dialogues, a proposed new model for a negotiation agent, and automatic and human evaluation experiments that demonstrate superior performance over several baselines.  The reviewers especially appreciate the value of the newly created dataset and methodology, and is seen as the paper's main asset.  They also acknowledge the comprehensive evaluations and positive results in terms of both objective and subjective metrics.  On the other hand, the reviewers are less excited about the proposed approach, considering it to have only limited novelty.  The paper could be improved by pitching it more as a resource paper.

---

### Decision · Program_Chairs · 2023-10-07

**Decision:**

Accept-Findings

**Comment:**

This paper contributes a valuable new dataset of negotiation dialogues, a proposed new model for a negotiation agent, and automatic and human evaluation experiments that demonstrate superior performance over several baselines.  The reviewers especially appreciate the value of the newly created dataset and methodology, and is seen as the paper's main asset.  They also acknowledge the comprehensive evaluations and positive results in terms of both objective and subjective metrics.  On the other hand, the reviewers are less excited about the proposed approach, considering it to have only limited novelty.  The paper could be improved by pitching it more as a resource paper.